# Abilities of Rare Sugar Members to Release Glucagon-like Peptide-1 and Suppress Food Intake in Mice

**DOI:** 10.3390/nu17071221

**Published:** 2025-03-31

**Authors:** Yuta Masuda, Kento Ohbayashi, Kengo Iba, Rika Kitano, Tomonori Kimura, Takako Yamada, Tohru Hira, Toshihiko Yada, Yusaku Iwasaki

**Affiliations:** 1Laboratory of Animal Science, Graduate School of Life and Environmental Sciences, Kyoto Prefectural University, Kyoto 606-8522, Japan; masuda-19@kpu.ac.jp (Y.M.); k.ohbayashi.9813@gmail.com (K.O.);; 2Research and Development, Matsutani Chemical Industry Company, Limited, Itami 664-8508, Japan; 3Laboratory of Nutritional Biochemistry, Research Faculty of Agriculture, Hokkaido University, Sapporo 060-8589, Japan; hira@agr.hokudai.ac.jp; 4Center for Integrative Physiology, Kansai Electric Power Medical Research Institute, Osaka 553-0003, Japan; 5Department of Diabetes, Endocrinology and Metabolism/Rheumatology and Clinical Immunology, Graduate School of Medicine, Gifu University, Gifu 501-1194, Japan; 6Center for One Medicine Innovative Translational Research, Institute for Advanced Study, Gifu University, Gifu 501-1194, Japan

**Keywords:** rare sugars, food intake, GLP-1, allose, fructose, allulose, tagatose, sorbose

## Abstract

**Background/Objectives:** Rare sugars, which naturally exist in small quantities, have gained attention as next-generation functional sugars due to their sweetness and low calorie content. Some of them have already been commercialized. Rare sugar-containing syrups, produced through alkaline isomerization of high-fructose corn syrup, are effective in preventing obesity and type 2 diabetes. However, the mechanisms underlying these effects remain incompletely understood. Recently, D-allulose has been found to improve hyperphagic obesity by stimulating the secretion of the intestinal hormone glucagon-like peptide-1 (GLP-1). The present study aimed to determine the comparative effects of aldohexoses (D-glucose, D-allose) and ketohexoses (D-fructose, D-allulose, D-tagatose, D-sorbose) on GLP-1 secretion and food intake in male mice. **Method and Results:** Single peroral administration of four ketohexoses at 1 and 3 g/kg, but not aldohexoses at 1 and 3 g/kg, significantly increased plasma GLP-1 concentrations with comparable efficacy. Moreover, these ketohexoses at 1 g/kg suppressed food intake in the short term, an effect blunted by GLP-1 receptor antagonism. In contrast, zero-calorie D-allose at 3 g/kg suppressed feeding without raising plasma GLP-1 levels. **Conclusions:** These results demonstrate that D-allulose, D-tagatose, and D-sorbose, which are low-calorie rare sugars classified as ketohexoses, suppress food intake through promoting GLP-1 secretion, showing their potential to prevent and/or ameliorate type 2 diabetes, obesity and related diseases.

## 1. Introduction

Rare sugars are defined as monosaccharides and their derivatives that exist in small quantities in nature. Although there are more than 50 kinds of rare sugars, their presence on earth is less than 0.1% [1]. In contrast, common sugars such as sucrose and its components, D-glucose and D-fructose, are abundant in nature and are widely used as sweeteners due to their energy content and sweet taste. These common sugars are easily overconsumed, and it is well known that excessive intake can lead to metabolic disorders such as obesity, diabetes, and dyslipidemia [2]. Many rare sugars possess sweetness while being low in calories because they are poorly metabolized in the body. Therefore, rare sugars are expected to contribute to the prevention of excessive energy intake as sweeteners and may play a role in preventing lifestyle-related diseases such as obesity and diabetes [3].

In recent years, the large-scale industrial synthesis of individual rare sugars has advanced, primarily through enzymatic methods [4,5,6,7,8]. Simultaneously, a chemical method, alkaline isomerization, has been established to produce multiple rare sugars from isomerized sugar. This process yields rare sugar syrup containing at least 13% rare sugars and is more cost-effective than enzymatic synthesis [9]. The rare sugar syrup is a mixed syrup containing 45% D-glucose, 29% D-fructose, 5% D-allulose, 5% D-sorbose, 2% D-tagatose, 1% D-allose, and 13% other sugars, among which D-allulose, D-sorbose, D-tagatose, and D-allose are classified as rare sugars. Numerous studies have reported the physiological functions of this rare sugar syrup, including its anti-obesity and anti-diabetic effects, in both animal and human studies [1,10,11]. However, the mechanisms underlying these effects remain to be fully revealed.

Glucagon-like peptide-1 (GLP-1) is a peptide hormone secreted from intestinal enteroendocrine L-cells, with its secretion being enhanced after meals. This endogenous GLP-1 has an extremely short half-life due to degradation by dipeptidyl peptidase-4 (DPP-4); however, it plays a crucial role in suppressing food intake and improving glucose tolerance by acting on vagal afferent nerves distributed near the gastrointestinal system [12]. We previously reported that the rare sugar D-allulose promotes GLP-1 release, thereby preventing hyperphagic obesity [13]. Similar anti-obesity effects are also observed with the rare sugar syrup containing various rare sugars, suggesting that other rare sugars besides D-allulose may also promote GLP-1 secretion and suppress food intake. However, the effects of D-sorbose, D-tagatose, and D-allose, components of rare sugar syrup, on GLP-1 secretion and feeding behavior remain unclear.

In this study, we focused on two aldohexoses (D-glucose, D-allose) and four ketohexoses (D-fructose, D-allulose, D-tagatose, D-sorbose) contained in the rare sugar syrup. We first examined the effect of these monosaccharides on plasma GLP-1 levels in overnight-fasted C57BL/6J male mice. Next, we investigated the effect of single-monosaccharide administration on feeding behavior in mice and the involvement of GLP-1 receptor signaling by pretreatment with a GLP-1 receptor antagonist. We found that D-allulose, D-tagatose, and D-sorbose, low- or zero-calorie sugars with sweet taste, promote GLP-1 secretion and thereby suppress food intake in the short term. In contrast, zero-calorie D-allose suppresses food intake without increasing plasma GLP-1 levels. These functional rare sugars hold promise as food ingredients that could help prevent and treat hyperphagic obesity, diabetes, and related diseases.

## 2. Materials & Methods

### 2.1. Materials

Of the two aldohexose and four ketohexoses used in this study (Figure 1), D-glucose and D-fructose were purchased from Fujifilm Wako Pure Chemical Corporation (Osaka, Japan). The other four rare sugars, D-allose, D-allulose, D-tagatose, and D-sorbose, were provided by Matsutani Chemical Industry Co., Ltd. (Itami, Japan), with purities exceeding 98%. The monosaccharides were dissolved in distilled water for administration to the mice. GLP-1 receptor antagonist exendin(9–39) (Ex(9–39)), of guaranteed reagent grade, was custom-synthesized by GenScript Japan (Tokyo, Japan). All the other chemicals were of guaranteed reagent grade or higher. 

### 2.2. Animals

Male C57BL/6J mice were purchased from Jackson Laboratory Japan, Inc. (Yokohama, Japan) and housed under controlled temperature (22.5 °C ± 2 °C), humidity (55% ± 10%), and light (light phase: 7:30–19:30). The purchased mice were acclimated to the new environment for at least one week before the experiments. Standard chow (CE-2, CLEA Japan, Tokyo, Japan) and water were available ad libitum to the mice. All experiments were performed on male mice between 8 and 20 weeks of age. The animal experiments were carried out after receiving approval from the Institutional Animal Experiment Committee of the Kyoto Prefectural University and in accordance with the Institutional Regulations for Animal Experiments (approval number: KPU030402-C, KPU060327-RC-5).

### 2.3. Measurement of Plasma GLP-1 Concentration in the Portal Vein

Mice were fasted overnight (18:00 to 10:00 the next day). The six monosaccharide solutions (1 g/10 mL/kg or 3 g/10 mL/kg) or saline were administered perorally (po) into the stomach using a stainless-steel feeding needle at 10:00. Regarding the dosage, our previous study on D-allulose identified 1 g/kg as the minimum effective dose for inducing GLP-1 secretion [13]. Therefore, we examined the effects of 1 and 3 g/kg doses in this study. Blood samples were collected from the portal vein under isoflurane anesthesia 1 h after injection. The sampling syringe contained heparin (final concentration; 50 IU/mL), aprotinin (final concentration; 500 KIU/mL), and DPP-4 inhibitor vildagliptin (final concentration; 10 μM). Plasma was collected after centrifugation (4000 rpm, 10 min at 4 °C) and stored at −80 °C until assay. Active GLP-1 levels were measured using active GLP-1 ELISA kits (EGLP-35K; Millipore, Burlington, MA, USA). We have confirmed that saline administration does not affect the GLP-1 concentration in portal vein plasma 1 h after administration, serving as a control (water: 23.01 ± 4.66 pM; saline: 24.64 ± 2.41 pM; means ± SEM, *n* = 4; not significant difference by unpaired *t*-test).

### 2.4. Measurements of Food Intake

Mice were housed in individual cages and sufficiently habituated to a standard powdered diet (CE-2, 3.4 kcal/g, CLEA Japan) in a feeding box (SN-950, Shinano Manufacturing Co., Ltd., Tokyo, Japan) and to being handled for at least 1 week before the experiments. Mice were deprived of food from 18:00 with free access to water 1 day before the experiment (16 h fasting). On the next day at 9:50, monosaccharide solutions (1 or 3 g/10 mL/kg) or saline (10 mL/kg) were administered po, and then CE-2 powdered diet was given at 10:00. The feeding box, including powdered food and food spillage, was weighed at 1, 3, 6, and 24 h after starting the test. The cumulative energy intake, including food eaten (CE-2; 3.4 kcal/g) and administered monosaccharides (D-glucose and D-fructose; 4 kcal/g, D-tagatose; 2 kcal/g, D-sorbose; 1 kcal/g, D-allose and D-allulose; 0 kcal/g) at each time, was expressed. We confirmed that a single po administration of saline did not significantly affect food intake compared to water, serving as a control (0.31 ± 0.03 g at 1 h and 1.09 ± 0.04 g at 3 h after water administration; 0.30 ± 0.06 g at 1 h and 0.98 ± 0.14 g at 3 h after saline administration; means ± SEM, *n* = 6; not significant difference by unpaired *t*-test).

To examine the involvement of GLP-1R signaling, Ex(9–39) at 600 nmol/5 mL/kg or saline (5 mL/kg) was intraperitoneally (ip) administered with a single shot at 15 min before monosaccharides po administration. The dosage of this GLP-1 receptor antagonist was based on our previous research, where Ex(9–39) at 600 nmol/kg was shown to inhibit the anorexigenic effect of D-allulose [13]. Our previous study demonstrated that pretreatment with Ex(9–39) alone does not affect food intake in mice [13].

### 2.5. Statistical Analysis

All data are shown as means ± SEM. Statistical analysis was performed using one-way ANOVA or the Kruskal–Wallis test as multiple comparisons, and the Mann–Whitney test for the two-sample comparisons. When ANOVA indicated significant differences among groups, post hoc comparisons were conducted using Tukey’s test, Dunnett’s test, or Dunn’s test. For the analysis of relative values, nonparametric tests such as the Kruskal–Wallis test or the Mann–Whitney test were used. All statistical analyses were performed using Prism 10 (GraphPad Software, San Diego, CA, USA), and *p* < 0.05 was considered statistically significant.

## 3. Results

### 3.1. Single Peroral Administration of Ketohexoses, but Not Aldohexose, Elevates Plasma GLP-1 Concentrations in a Dose-Dependent Manner

We investigated the ability of two aldohexoses (D-glucose and D-allose, Figure 1) and four ketohexoses (D-fructose, D-allulose, D-tagatose, and D-sorbose, Figure 1) to increase active GLP-1 concentrations in the portal vein after peroral (po) administration. We previously reported that D-allulose increases plasma GLP-1 concentrations, peaking at 1 h after administration [13]. Based on this, GLP-1 concentrations were measured in the portal vein plasma 1 h after administration.

Po administration of the four ketohexoses at 1 g/kg (D-fructose, D-allulose, D-tagatose, and D-sorbose) significantly increased active GLP-1 concentrations in portal vein by approximately 3–4-fold compared to saline (Saline: 9.22 ± 1.02 pM vs. D-fructose: 37.6 ± 7.20 pM, D-allulose: 35.8 ± 5.52 pM, D-tagatose: 44.1 ± 4.09 pM, D-sorbose: 31.6 ± 4.65 pM; Figure 2A). At a higher dose of 3 g/kg, these ketohexoses further increased plasma GLP-1 concentrations by approximately 4–6-fold compared to saline (Saline: 15.2 ± 1.75 pM vs. D-fructose: 66.72 ± 8.40 pM, D-allulose: 92.1 ± 10.7 pM, D-tagatose: 75.0 ± 9.82 pM, D-sorbose: 94.8 ± 9.83 pM; Figure 2B). There were no significant differences in the ability to stimulate GLP-1 secretion among the four ketohexoses at each administered dose (Figure 2A,B). On the other hand, neither of the two aldohexoses (D-glucose and D-allose) at either 1 or 3 g/kg significantly increased plasma GLP-1 concentrations (Figure 2A,B), although D-allose at 3 g/kg showed a tendency to increase GLP-1 concentrations compared to saline (Figure 2B).

### 3.2. Po Administration of Four Ketohexoses and D-Allose, but Not D-Glucose, Results in Short-Term Suppression of Food Intake

The enhancement of GLP-1 secretion from the gut by food or D-allulose is known to contribute to the short-term suppression of food intake [13,14]. Therefore, we investigated the effects of two aldohexoses and four ketohexoses on feeding behavior. Monosaccharides were administered perorally immediately before refeeding to mice fasted overnight, and food intake was measured at 1, 3, 6, and 24 h after administration. In Figure 3, food intake for each monosaccharide experiment is shown as absolute values (cumulative intake, kcal), while in Figure 4, the results are presented as relative values (relative cumulative food intake, % to saline group), normalized based on the saline group for each time point on each experiment day.

A single po administration of D-glucose at 1 and 3 g/kg did not alter food intake at any time point (Figure 3A and Figure 4). Unexpectedly, D-allose, which did not significantly increase GLP-1 secretion, significantly reduced food intake, particularly at 3 g/kg, during the 1 to 6 h period after administration (Figure 3B and Figure 4). The four ketohexoses (D-fructose, D-allulose, D-tagatose, and D-sorbose), which significantly stimulated GLP-1 secretion at 1 and 3 g/kg, reduced food intake in a dose-dependent manner (Figure 3C–F and Figure 4). These anorexigenic effects were significant for up to 6 h after administration, particularly at the 3 g/kg dose (Figure 4), but they ceased by 24 h. The anorexigenic effects of the four ketohexoses were comparable (Figure 4).

### 3.3. GLP-1 Receptor Antagonism Blunts the Anorexigenic Effects of Ketohexoses but Not That of D-Allose

We showed the anorexigenic effects of four ketohexoses and D-allose in this study. To examine the role of GLP-1 receptor signaling in these effects, we pretreated mice with the GLP-1 receptor antagonist Ex(9–39). Ex(9–39) at 600 nmol/kg was administered intraperitoneally (ip) 15 min before po administration of ketohexoses (D-fructose, D-allulose, D-tagatose, and D-sorbose; 1 g/kg), and food intake was measured over time after refeeding. As shown in Figure 5A–C, pretreatment with Ex(9–39) did not significantly reduce food intake in the four ketohexose groups compared to the saline group at any time point. Additionally, as shown in Figure 5D–F, which was reanalyzed to compare Figure 4 and Figure 5A–C, pretreatment with Ex(9–39) significantly attenuated, or tended to attenuate, the anorexigenic effects of ketohexoses, particularly at 1 and 6 h after administration.

In contrast, similar Ex(9–39) pretreatment failed to inhibit the anorexigenic effect of D-allose at 3 g/kg, particularly at 3 and 6 h after administration (Figure 5A–C). Moreover, no significant difference in food intake was observed between the D-allose group with or without Ex(9–39) pretreatment (Figure 5D–F).

## 4. Discussion

In this study, we investigated the effects of four ketohexoses (D-fructose, D-allulose, D-tagatose, and D-sorbose) and two aldohexoses (D-glucose and D-allose) on GLP-1 secretion and feeding behavior in mice. All four ketohexoses significantly increased blood GLP-1 levels and short-term suppressed food intake immediately after a single po administration. Furthermore, the anorexigenic effects of these ketohexoses were blunted by pretreatment with GLP-1 receptor antagonist. In contrast, the aldohexoses D-glucose and D-allose did not significantly alter blood GLP-1 levels. Unexpectedly, po administration of D-glucose did not affect feeding behavior in mice, even though it is known to increase blood glucose levels. On the other hand, D-allose reduced food intake, similarly to the ketohexoses mentioned above. This anorexigenic effect of D-allose was not influenced by GLP-1 receptor antagonist pretreatment, suggesting that D-allose suppresses feeding independently of GLP-1 receptor signaling. These findings demonstrate that low/zero-calorie rare sugars, specifically ketohexoses, exert anorexigenic effects via GLP-1 secretion and GLP-1 receptor signaling.

This study identified GLP-1 release by all four ketohexoses. D-Fructose and D-allulose are known as GLP-1 releasers in rodents and humans [15,16]. This study found that D-sorbose and D-tagatose are novel GLP-1 releasers in mice. Further research is needed to determine whether these newly identified GLP-1 releasers are effective in humans as well as in rodents.

This study demonstrates that the chemical structure of ketohexoses plays an important role in GLP-1 secretion. However, the molecular mechanism of GLP-1 secretion induced by ketohexoses remains unclear. SGLT1 (sodium-glucose cotransporter 1) is known to be involved in GLP-1 secretion, but ketohexoses have a low affinity for SGLT1 [17,18]. Instead, ketohexoses have high affinity for GLUT5 (glucose transporter 5), which plays a critical role in the absorption of D-fructose and D-allulose [17,18]. Previous reports have shown that GLUT5 is important for GLP-1 secretion by D-fructose and D-allulose [15,17]. Therefore, GLUT5 could be involved in the GLP-1 release by ketohexoses. The GLP-1 secretion induced by ketohexoses may be influenced by intraluminal osmotic and distension stimuli generated by their slow absorption rate. High osmotic pressure in the intestinal lumen can increase distension stimuli associated with luminal expansion, thereby activating vagal afferent nerves that express GLP-1 receptors [19]. In support of this, intestinal distension induced by administering a carbonated solution containing pectin to mice has been shown to promote GLP-1 secretion [20]. A recent study demonstrated that the GLP-1 secretion promoted by D-allulose is due to intestinal distension caused by its slow absorption rate [21]. Ketohexoses exemplified by D-fructose are absorbed more slowly than D-glucose [22]. Therefore, the mechanism underlying the GLP-1 secretion observed in this study may involve intestinal distension caused by the slower absorption of ketohexoses.

In this study, a single po administration of the aldohexoses D-glucose and D-allose in mice did not result in significant changes in plasma GLP-1 levels after one hour. In humans, D-glucose intake has been widely reported to significantly increase plasma GLP-1 levels [15]. However, in our studies using mice and rats, GLP-1 responses to D-glucose have not been observed [13,17]. A previous report has highlighted the challenge of measuring plasma GLP-1 levels in mice [23], which may have contributed to our inability to detect aldose-induced GLP-1 secretion in this study. In mice, GLP-1 undergoes pronounced additional endoproteolytic degradation by neprilysin (neutral endopeptidase 24.11, NEP), which may hinder its detection using ELISA kits [24]. Therefore, it is presumed that the GLP-1 secretory response to aldoses is too weak to be detected in mice. On the other hand, it has been reported that oral administration of high doses of glucose (3–5 g/kg) in mice significantly increases plasma GLP-1 levels within 5–10 min after administration [15,25,26]. Future studies should focus on analyzing the temporal variations in GLP-1 levels to further elucidate the differences in GLP-1 secretory responses between aldohexoses and ketohexoses. Moreover, evaluating the effects of D-allose on GLP-1 secretion in humans will be an important area for further investigation.

There have been many reports suggesting that rare sugars contribute to the suppression of food intake and/or enhanced satiation. For example, D-allulose has been shown to reduce food intake in rodents [13] and enhance satiation in humans [16]. D-sorbose reportedly exerts long-term anorexigenic effects in rats when incorporated into their diet [27]. Similarly, D-tagatose has been reported to exert short-term anorexigenic effects in humans [28]. On the other hand, the effect of D-fructose on food intake remains controversial, as studies report both reductions [29,30] and no changes [31,32]. The present study demonstrates that oral administration of four ketohexoses, including D-fructose, D-allulose, D-sorbose, and D-tagatose, as well as the aldohexose D-allose, suppresses food intake. Notably, we are the first to report that D-allose exerts an anorexigenic effect. Furthermore, our findings are the first to demonstrate that GLP-1 receptor signaling serves as a key mediator of the suppression of food intake by ketohexoses, excluding D-allulose [13]. Remarkably, D-allose, an aldohexose, reduces food intake through a pathway independent of GLP-1 receptor signaling. However, the underlying mechanism remains to be determined. Ketohexoses such as D-fructose and D-allulose have limited permeability across the blood–brain barrier, making it unlikely that they directly affect the central nervous system [33]. In contrast, aldohexoses like D-glucose and D-allose exhibit greater permeability [33,34]. Therefore, D-allose may directly act on the central nervous system through a mechanism that does not require GLP-1 receptor signaling.

Excessive consumption of sugars, particularly glucose and fructose, has been linked to an increased risk of cardiovascular disease, diabetes, and obesity [35,36]. In contrast, many rare sugars have gained attention as low-energy sugars that are metabolized less efficiently than glucose and fructose. Research on allulose is relatively advanced, and its high safety has been demonstrated in studies where individuals, including those with borderline and type 2 diabetes, consumed 15 g of D-allulose daily for 12 weeks [1,37]. Based on such findings, allulose has been granted Generally Recognized as Safe (GRAS) status by the U.S. Food and Drug Administration (FDA) (Notice 693). In this study, we focused on four types of rare sugars that can be industrially synthesized in large quantities and evaluated their short-term benefits using male mice. All rare sugars examined demonstrated beneficial effects, either by stimulating GLP-1 secretion, enhancing satiety, or both. Building on these findings, future studies should explore the long-term benefits and safety of these rare sugars, transitioning from animal studies to human research.

## 5. Conclusions

The present study identifies D-sorbose (1 kcal/g) and D-tagatose (2 kcal/g) as novel low-calorie rare sugars that suppress food intake through GLP-1 secretion. Furthermore, this study reveals the anorexigenic effect of D-allose, a zero-calorie rare sugar. With the global rise in obesity and diabetes, the excessive use of caloric sweeteners has become a significant societal concern. Despite their low- or zero-caloric content, these rare sugars promote GLP-1 secretion and induce satiation, similarly to caloric nutrients. Therefore, they hold potential as functional food ingredients for preventing overeating and promoting glycemic control.

## Figures and Tables

**Figure 1 nutrients-17-01221-f001:**
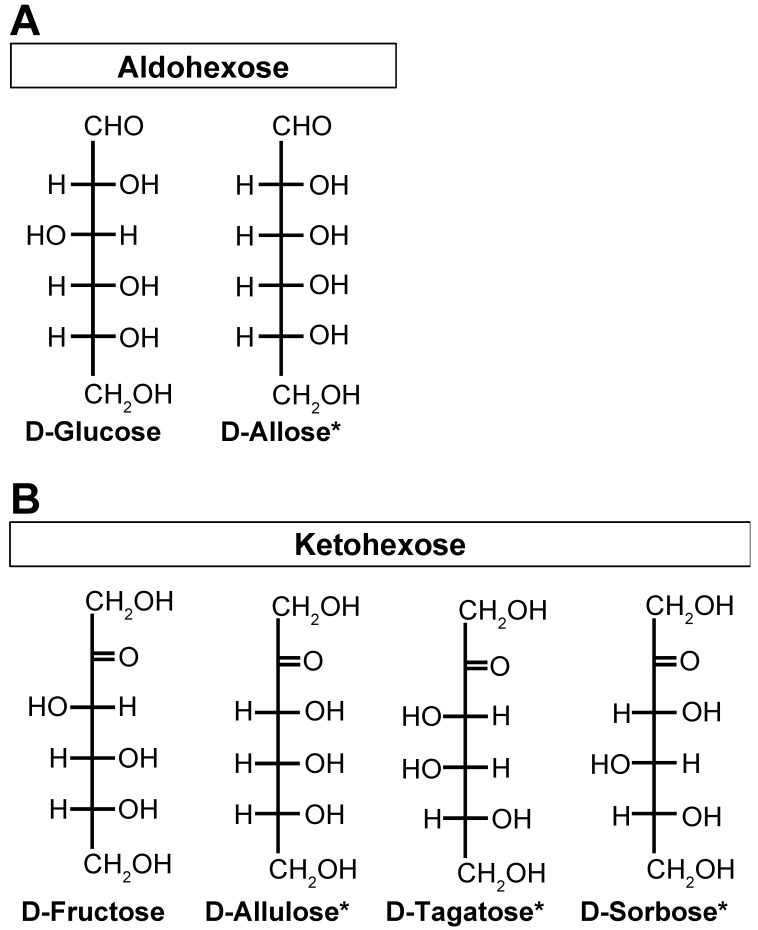
Chemical structures of the monosaccharides/hexoses investigated in this study: four rare sugars and two nonrare sugars. D-glucose and D-allose are aldohexoses that contain an aldehyde group (**A**). D-fructose, D-allulose, D-tagatose, and D-sorbose are ketohexoses that contain a ketone group (**B**). Of these, D-allose, D-allulose, D-tagatose, and D-sorbose are rare sugars, indicated by an asterisk (*).

**Figure 2 nutrients-17-01221-f002:**
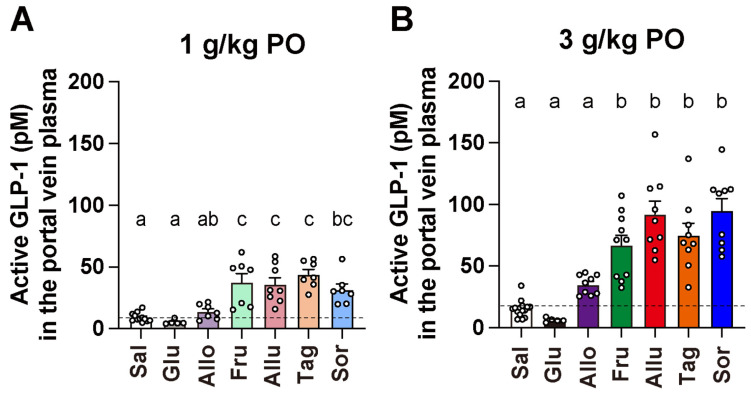
Comparison of plasma active GLP-1 concentrations after administration of six monosaccharides: All ketohexoses, except aldohexoses, promote GLP-1 secretion. Overnight-fasted C57BL/6J mice received monosaccharide solutions at either 1 g/kg (**A**) or 3 g/kg (**B**) via a stainless-steel feeding needle directly into the stomach. Blood samples were collected from the portal vein 1 h after administration, and plasma active GLP-1 concentrations were measured. The dashed lines in (**A**,**B**) represent the mean value of the saline group. *n* = 5–15. Different letters indicate significant differences with *p* < 0.05 determined by one-way ANOVA followed by Tukey’s test. GLP-1: glucagon-like peptide-1. Sal: saline. Glu: D-glucose. Allo: D-allose. Fru: D-fructose. Allu: D-allulose. Tag: D-tagatose. Sor: D-sorbose.

**Figure 3 nutrients-17-01221-f003:**
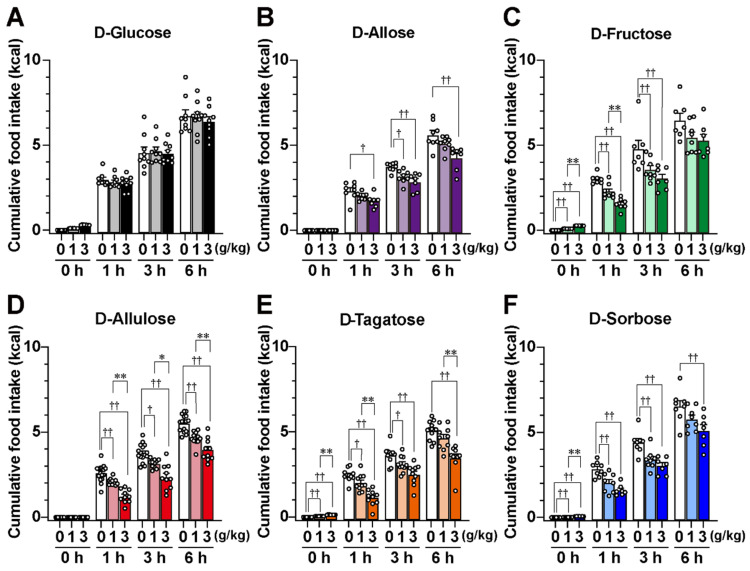
Single peroral administration of rare sugars or ketohexoses suppresses food intake. Mice were fasted overnight (16 h) and then perorally administered either saline or a monosaccharide solution ((**A**) D-glucose, (**B**) D-allose, (**C**) D-fructose, (**D**) D-allulose, (**E**) D-tagatose, (**F**) D-sorbose) at doses of 1 g/kg or 3 g/kg. The mice were fed immediately after administration, and cumulative food intake was measured at 1 h, 3 h, and 6 h after administration. The energy values of the administered solutions are indicated at 0 h in each figure; cumulative food intake includes the energy from the administered monosaccharides. *n* = 7–16. † *p* < 0.05, †† *p* < 0.01 by one-way ANOVA followed by Dunnett’s test compared to the saline po group. * *p* < 0.05, ** *p* < 0.01 by one-way ANOVA followed by Tukey’s test for comparisons between 1 g/kg and 3 g/kg.

**Figure 4 nutrients-17-01221-f004:**
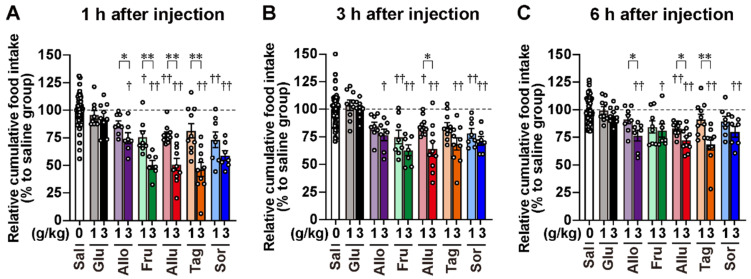
Comparison of the anorexigenic effects of monosaccharides. The data on cumulative food intake after administration of the six monosaccharides shown in Figure 3 were reanalyzed as relative cumulative food intake at 1 h (**A**), 3 h (**B**), or 6 h (**C**) after injection. The saline group was included in all experiments conducted on different days. Therefore, the cumulative food intake after monosaccharide administration was normalized to the saline group results from the same day. The dashed lines in each figure indicate the average value of the saline group. *n* = 104 for the saline group and *n* = 7–11 for other groups. † *p* < 0.05, †† *p* < 0.01 by the Kruskal–Wallis test followed by Dunn’s test vs. the saline group. * *p* < 0.05, ** *p* < 0.01 by the Mann–Whitney test. Sal: saline, Glu: D-glucose. Allo: D-allose. Fru: D-fructose. Allu: D-allulose. Tag: D-tagatose. Sor: D-sorbose.

**Figure 5 nutrients-17-01221-f005:**
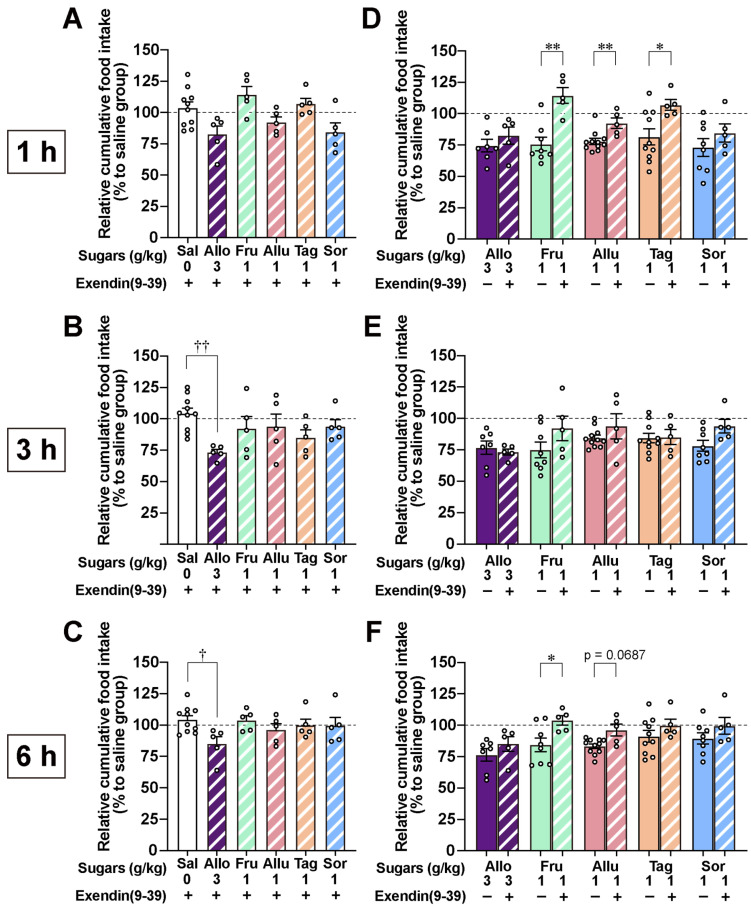
Pretreatment with GLP-1 receptor antagonist exendin(9–39) blunts the anorexigenic effects of ketohexose administration but not those of D-allose. (**A**–**C**) Exendin(9–39) at 600 nmol/kg was administered intraperitoneally to overnight-fasted mice 15 min before administration of either saline or monosaccharides (3 g/kg: D-allose; 1 g/kg: D-fructose, D-allulose, D-tagatose, and D-sorbose). Cumulative food intake was measured at 1 h (**A**), 3 h (**B**), and 6 h (**C**) after administration. These experiments were performed over multiple days; therefore, the results are presented as relative cumulative food intake (% kcal relative to the saline group). (**D**–**F**) To compare the effects of exendin(9–39), the results from Figure 5A–C and Figure 4 were reanalyzed for each time point. † *p* < 0.05, †† *p* < 0.01 by Kruskal–Wallis test followed by Dunn’s test vs. saline group. * *p* < 0.05, ** *p* < 0.01, and *p* = 0.068 by Mann–Whitney test between the two groups. Sal: saline, Glu: D-glucose. Allo: D-allose. Fru: D-fructose. Allu: D-allulose. Tag: D-tagatose. Sor: D-sorbose.

## Data Availability

The original contributions presented in this study are included in the article. Further inquiries can be directed to the corresponding authors.

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
