# Peer review of "Abilities of Rare Sugar Members to Release Glucagon-like Peptide-1 and Suppress Food Intake in Mice"

_nutrients, 2025, doi:10.3390/nu17071221_

Round 1
Reviewer 1 Report
Comments and Suggestions for Authors
In the manuscript, Masuda et al. present a study on the effects of rare sugar members on glucagon-like peptide-1 (GLP-1) secretion and their subsequent impact on food intake in mice. The authors provide a thorough investigation into the comparative effects of aldohexoses (D-glucose, D-allose) and ketohexoses (D-fructose, D-allulose, D-tagatose, D-sorbose) on GLP-1 release and food intake. The study is well-structured and contributes valuable insights into the impacts of rare sugars. However, there are a few concerns to be addressed:
Major concerns:
- If the sugars were dissolved in distilled water, was saline a proper control?
- Figure 5, In the experiment testing the role of GLP-1 receptor signaling in the anorexigenic effects of ketohexoses and D-allose, why the authors used 1g/kg of ketohexoses but not 3g/kg as the D-allose? As shown in Figure 3, 1g/kg of Fructose or tagatose did not consistently reduce food intake 3, 6 hr after po administration, the authors should use 3g/kg of ketohexoses instead of 1g/kg in figure 5.
Minor concerns:
- In results 3.1, the structures of six sugars were shown in Figure 1 not Table 1.
- Figure 5D-5F, all the values were normalized to saline group, was the saline group with or without Exendin (9-39) pretreatment, the authors should clarify.
Author Response
Reviwer#1
Comments and Suggestions for Authors
In the manuscript, Masuda et al. present a study on the effects of rare sugar members on glucagon-like peptide-1 (GLP-1) secretion and their subsequent impact on food intake in mice. The authors provide a thorough investigation into the comparative effects of aldohexoses (D-glucose, D-allose) and ketohexoses (D-fructose, D-allulose, D-tagatose, D-sorbose) on GLP-1 release and food intake. The study is well-structured and contributes valuable insights into the impacts of rare sugars. However, there are a few concerns to be addressed:
Major concerns:
1) If the sugars were dissolved in distilled water, was saline a proper control?
Response to major comment 1
Thank you for your important question. We chose saline as the control because it is an isotonic solution and is expected to have minimal physiological effects. To confirm this, we examined the effects of water and saline on plasma GLP-1 levels and feeding behavior. Our results showed no significant differences between the two groups. A single oral administration of water or saline (10 ml/kg) did not lead to any difference in total GLP-1 levels in portal vein plasma at 1 hour post-administration (water: 23.01 ± 4.66 pM; saline: 24.64 ± 2.41 pM; means ± SEM, n = 4; not significant difference by unpaired t-test). Similarly, no significant difference in food intake was observed between the two groups after a single oral administration of water or saline (10 ml/kg) during the nocturnal feeding period (0.31 ± 0.03 g at 1 h and 1.09 ± 0.04 g at 3 h after water administration; 0.30 ± 0.06 g at 1 h and 0.98 ± 0.14 g at 3 h after saline administration; means ± SEM, n = 6; not significant difference by unpaired t-test). The rationale for using saline as the control is described in the Methods section (Page 7, Line 156-158; Page 7, Line 171-173).
2) Figure 5, In the experiment testing the role of GLP-1 receptor signaling in the anorexigenic effects of ketohexoses and D-allose, why the authors used 1g/kg of ketohexoses but not 3g/kg as the D-allose? As shown in Figure 3, 1g/kg of Fructose or tagatose did not consistently reduce food intake 3, 6 hr after po administration, the authors should use 3g/kg of ketohexoses instead of 1g/kg in figure 5.
Response to major comment 2
Thank you for your valuable comments. We selected a dose of 1 g/kg to examine the involvement of GLP-1R for the following two reasons: (1) at higher concentrations of rare sugars, GLP-1R antagonist Ex(9-39) may not exert sufficient inhibitory effects, and (2) both tagatose and fructose showed significant or trend-level suppression of food intake at 1 and 3 hours post-administration.
(1) In our previous study, the anorexigenic effect of 3 g/kg PO allulose was completely abolished in GLP-1R knockout (GLP-1RKO) mice but was not entirely eliminated by pretreatment with the GLP-1R antagonist Ex(9-39) (PMID29317623, Fig. 2). In contrast, the anorexigenic effect of 1 g/kg PO allulose was significantly attenuated by GLP-1R antagonist pretreatment (PMID29317623, Fig. 2). Our previous report has been suggested that GLP-1 secreted in the gut in response to allulose acts on GLP-1Rs expressed on vagal sensory nerves located near the enteroendocrine cells, thereby suppressing food intake (PMID29317623). Since it is difficult for an intraperitoneally administered GLP-1R antagonist to reach concentrations sufficient to completely inhibit local GLP-1/GLP-1R interactions in the gut, we used the minimal effective dose of 1 g/kg to evaluate the effect of GLP-1R antagonism in the present study.
(2) Regarding Figure 3, re-evaluation using Dunnett’s test for comparisons with the control group confirmed that both 1 g/kg fructose and 1 g/kg tagatose significantly suppressed food intake at 1 and 3 hours post-administration (Fig. 3). In Figure 4, pretreatment with Ex(9-39) attenuated the anorexigenic effects of all four ketohexoses at all time points (Fig. 5A–C). Additionally, the anorexigenic effects of 1 g/kg fructose, allulose, and tagatose were significantly reduced by Ex(9-39) treatment compared to the untreated condition (Fig. 5D). For sorbose, a comparison between Figures 4A and 5A suggests that its anorexigenic effect was also significantly attenuated by Ex(9-39) pretreatment. These findings indicate the involvement of GLP-1R in the anorexigenic effects of the four ketohexoses.
In the revised manuscript, we have added a description of the statistical analysis using Dunnett’s test for comparisons with the control group in Figure 3 (Fig. 3; Page 21, Lines 572–574).
Minor concerns:
1) In results 3.1, the structures of six sugars were shown in Figure 1 not Table 1.
Response to minor comment 1
I sincerely apologize for the typographical error. The relevant section has been corrected (Page 6, Line 124; Page 8, Line 195 and 196).
2) Figure 5D-5F, all the values were normalized to saline group, was the saline group with or without Exendin (9-39) pretreatment, the authors should clarify.
Response to minor comment 2
Thank you for your important comment. In our previous study, neither saline nor Ex (9-39) pretreatment affected food intake in mice (PMID29317623). Therefore, in this study, we did not include saline pretreatment as a control experiment for Ex (9-39) pretreatment. To clarify this point, we have added the following statement to the relevant section in the Methods (Page 7, Lines 178–179).
Reviewer 2 Report
Comments and Suggestions for Authors
The paper “Abilities of rare sugar members to release glucagon-like peptide-1 and suppress food intake in mice” aimed to determine the comparative effects of aldohexoses (D-glucose, D-allose) and ketohexoses (D-fructose, D-allulose, D-tagatose, D-sorbose) on GLP-1 secretion and food intake in male mice. The research possesses significant implications and substantial application value, but there are still some issues that need to be addressed.
Comments:
Q1. The repetition rate of the manuscript is too high. It is suggested to further reduce it.
Q2. The description of the results in the abstract is insufficient. It is suggested that more data content be supplemented.
Q3. Only male mice were used in this study, and female mice were not involved. Whether gender differences affect carbohydrate metabolism and GLP-1 secretion is suggested to supplement the experimental data of female mice or related background information for discussion.
Q4. Two monosaccharide doses of 1g/kg and 3g/kg were set in the experiment, how was the dose range determined? What is the basis? Whether there are relevant references to support.
Q5. When studying the effect of GLP-1 receptor antagonists on the anorexigenic effect of saccharide, only saline controls were set, with or without an identified positive control drug, and the action through GLP-1 receptor signaling pathway was determined.
Q6. No comprehensive assessment of the health status of the mice was mentioned during the experiment, such as liver and kidney functions, blood routine and other indicators. Long-term intake of different saccharides may have potential effects on the health of mice. As some saccharides have been established to increase the risk of cardiovascular and cerebrovascular diseases in humans, the effects of saccharides in this paper need to be further discussed.
Author Response
Reviewer #2
The paper “Abilities of rare sugar members to release glucagon-like peptide-1 and suppress food intake in mice” aimed to determine the comparative effects of aldohexoses (D-glucose, D-allose) and ketohexoses (D-fructose, D-allulose, D-tagatose, D-sorbose) on GLP-1 secretion and food intake in male mice. The research possesses significant implications and substantial application value, but there are still some issues that need to be addressed.
Comments:
Q1. The repetition rate of the manuscript is too high. It is suggested to further reduce it.
Response to Q1
Thank you for your important comment. There was some overlap between the Results and Discussion sections, so we have removed the redundant parts from the Results section (Page 8, Lines 213–215; Page 9, Lines 237–238 and 252–254; Page 10, Lines 258–260).
Q2. The description of the results in the abstract is insufficient. It is suggested that more data content be supplemented.
Response to Q2
Thank you for your comment. Due to the word limit in the abstract, we have added information on the dosage.
Q3. Only male mice were used in this study, and female mice were not involved. Whether gender differences affect carbohydrate metabolism and GLP-1 secretion is suggested to supplement the experimental data of female mice or related background information for discussion.
Response to Q3
Thank you for pointing out this important issue. Our preliminary findings suggest that, similar to males, allulose also promotes GLP-1 secretion and reduces food intake in female mice (data not shown). While analysis in female mice is indeed important, we have chosen to first present our findings in male mice in this study. We plan to investigate this further in female mice in future research.
Q4. Two monosaccharide doses of 1g/kg and 3g/kg were set in the experiment, how was the dose range determined? What is the basis? Whether there are relevant references to support.
Response to Q4
Regarding the dosage, our previous study on D-allulose identified 1 g/kg as the minimum effective dose for inducing GLP-1 secretion and its anorexigenic effect (PMID29317623). Therefore, we examined the effects of 1 and 3 g/kg doses in this study. We have added this rationale to the relevant section in the Methods (Page 7, Lines 148–151).
Q5. When studying the effect of GLP-1 receptor antagonists on the anorexigenic effect of saccharide, only saline controls were set, with or without an identified positive control drug, and the action through GLP-1 receptor signaling pathway was determined.
Response to Q5
Thank you for your important comment. In our previous study, neither saline nor Ex (9-39) pretreatment affected food intake in mice (PMID29317623). Therefore, in this study, we did not include saline pretreatment as a control experiment for Ex (9-39) pretreatment. To clarify this point, we have added the following statement to the relevant section in the Methods (Page 7, Lines 178–179).
Q6. No comprehensive assessment of the health status of the mice was mentioned during the experiment, such as liver and kidney functions, blood routine and other indicators. Long-term intake of different saccharides may have potential effects on the health of mice. As some saccharides have been established to increase the risk of cardiovascular and cerebrovascular diseases in humans, the effects of saccharides in this paper need to be further discussed.
Response to Q6
Thank you for your important comment. We have added the relevant content to the final paragraph of the Discussion (Page 12, Lines 338–350).
Round 2
Reviewer 2 Report
Comments and Suggestions for Authors
No additional comments.